# Heart Rate Variability Control Using a Biofeedback and Wearable System

**DOI:** 10.3390/s22197153

**Published:** 2022-09-21

**Authors:** Eduardo Viera, Hector Kaschel, Claudio Valencia

**Affiliations:** Universidad de Santiago de Chile, Av. Víctor Jara N° 3519, Estación Central, Región Metropolitana, Santiago 9170124, Chile

**Keywords:** heart rate, heart rate variability, biofeedback, instrumental variables, MRAC, control system

## Abstract

Heart rate variability is an important physiological parameter in medicine. This parameter is used as an indicator of physiological and psychological well-being and even of certain pathologies. Research on biofeedback integrates the fields of biological application (physiological behavior), system modeling, and automated control. This study proposes a new method for modeling and controlling heart rate variability as heart rate acceleration, a model expressed in the frequency domain. The model is obtained from excitation and response signals from heart rate variability, which through the instrumental variables method and the minimization of a cost function delivers a transfer function that represents the physiological phenomenon. This study also proposes the design of an adaptive controller using the reference model. The controller controls heart rate variability based on the light actuators designed here, generating a conditioned reflex that allows individuals to self-regulate their state through biofeedback, synchronizing this action to homeostasis. Modeling is conducted in a target population of middle-aged men who work as firefighters and forest firefighters. This study validates the proposed model, as well as the design of the controllers and actuators, through a simple experiment based on indoor cycling. This experiment has different segments, namely leaving inertia, non-controlled segment, and actively controlled segment.

## 1. Introduction

Heart rate variability (HRV) is a parameter widely used as an identifier of physiological and mental well-being in people [1]. This cardiac parameter is used in systems in the area of physical well-being, as well as in the monitoring of vital signs. The monitoring of physiological variables has been expanded with the use of mobile devices, allowing indices of physical well-being, stress, etc. to be established [2].

The use of HRV is widespread and is used in areas of medicine for the detection and recognition of stress [3,4]; cardiovascular health [5]; insomnia [6], and autoimmune diseases, such as ulcerative colitis [7].

This parameter is obtained from electrocardiograms (ECGs) measuring the duration of the R-R intervals [8]. Having the heart rate, the HRV measurement needs a time window (both in the time domain and in the frequency domain) for its estimation; this presents an inherent delay in its processing time [9]. This is a problem, as the responses in the feedback system must be tolerant of these [10] delays.

On the other hand, HRV control represented by an adequate response time implies a direct benefit in the reestablishment of the balance of the autonomic nervous system, which is associated with involuntary actions of the body, including pain and vasovagal syncope [11,12,13,14].

Currently, the search for self-control of physiological functions in the human body has led to the application of feedback on these variables in the individual. This is shown in the work that seeks to train the responses of individuals to changes in HRV [15]. This concept has also evolved by integrating automatic control [2,16].

However, the raised solutions do not yet solve a fundamental problem, the response time of the HRV. The principal efforts associated with time series processing have optimized the response to obtain this parameter [17,18]. Solving the self-control of HRV using biofeedback over continuous time [2] contributes to physical and psychological well-being and safety in individuals such as firefighters and forest firefighters, who are constantly subjected to environments with extreme and aggressive conditions. Having a consequence of fatigue, dizziness, weakness, fainting, and so on [11,19,20].

## 2. Related Work

Controlling HRV has an important meaning in the safety and well-being of individuals subjected to work under hostile conditions. Currently, it is a challenge to find a model of HRV as a system, either dynamically in a discrete or continuous way, and it requires control strategies and actuators that allow, through their actions, to train individuals so that they respond to an external stimulus achieving self-control HRV.

The HRV is used in several areas of knowledge as an indicator of metabolic function and stress levels, as shown in [5,21]. This parameter was shown to be correlated with insomnia using polysomnographic studies that correlated changes in HRV with this pathology [6]. Similarly, this parameter is correlated with stress and mental health, as shown in [7,22,23,24].

From a cardiovascular point of view, HRV is currently used to correlate its behavior with severe and chronic heart disease [4] In the area of health wellness, the use of HRV is associated with physical performance metrics, which it is used for forms of physical training [25], and even behavioral training is associated with the controllability of this parameter [26].

In addition, the HRV has been used to determine the cardiac status of people affected by the COVID-19 virus [27]. The latter shows and supports HRV as a variable to control, has great challenges, and new potential areas of knowledge based on its control.

In feedback systems, the dynamic representation of the system under study is essential to achieve this goal. However, the differential representation of a physiological system is a really complex task, as models will always be subject to nonlinearities [28,29]. In this context, the identification of systems through the observation of an experiment to correlate variables that are not directly related but that are integrated by the principle of causality makes the use of instrumental variables (IV) an interesting method to apply in the estimation of a dynamic model of the HRV based on experimental observations, as a mechanical system, for example [30,31,32,33].

Considering controlling HRV can be considered as an input to a biofeedback system. Therefore, it makes it possible to perform actions that control the behavior of this variable [26]. An example of this is shown in conventional medicine in patients with joint and muscle pain problems [34,35,36], where biofeedback techniques generate self-correction of body positions that benefit study subjects.

This physiological and psychological control effect has been applied in the area of video games [37], also in physical training, training people to improve breathing, and even self-control of HRV in physical activities [15,38,39,40]. In the area of medicine, applications are being studied in anorectal recovery therapies, integrating the IoMT (Internet of Medical Things) concept [41,42].

Moreover, the use of biofeedback has proven results in the medical area with applications in the control of HRV showing that through external excitations such as harmonic sounds it is possible to improve HRV, and thus the homeostasis or self-regulatory process of the organism [43], in this solution HRV and biofeedback are integrated to generate self-control of mental stress in drivers [3].

Referring to automatic control, some controller applications based on fuzzy logic have been applied in biofeedback, such as [16,44,45,46]. In addition, other types of applications associated with automatic control are associated with the control of sleep apnea and physical behavior in sports, among other related areas, controllers have been applied in continuous and discrete time [47,48,49].

Regarding control signals in a biofeedback system, light signals are signals that practically have no time delay compared to the HRV response. Considering the psychological effect of light on individuals that works associated with the study of the effects of light intensity and its flickering frequency has a direct effect on the states of alertness of individuals [50,51]. The same goes for effects on individual stress as evidenced in [52,53], and even in the circadian cycle [54].

Note that in this work different areas are involved regarding the control of physiological variables; therefore, we resume the main contributions as follows.
We propose a model identification method via a transfer function for HRV focused on a specific population of individuals using the method of instrumental variables.We propose the design of an MRAC controller (Model Reference Adaptive Controller) using the estimated HRV model.We describe the design of light actuators to establish a conditioned reflex.We propose an experiment to validate the HRV model obtained through biofeedback.


The objective of our study was to investigate the ability to control the viability of heart rhythm through biofeedback, generating conditioned reflexes based on the training of individuals responding to external stimuli, such as light signals. Our hypothesis is that the use of an HRV-focused biofeedback system allows HRV regulation.

## 3. Materials and Methods

This research focused on the search for a dynamic and controllable model based on biofeedback. This is to generate a controlled system that allows individuals to self-monitor their HRV.

This research used, as input data, segments of electrocardiograms of a specific population: firefighters and forest firefighters who are exposed to aggressive and extreme environmental conditions that make the process of homeostasis very dynamic and reflected in the behavior of HRV [19,55]. The characteristics of this studied population are given by a universe of 12 individuals who have the following characteristics:Age: 35–45 years.Weight: 80–95 kg.Height: 160–181 cm.Occupation firefighters and forest firefighters.No heart disease.Physically active.

This research was approved by the Institutional Ethics Committee of the University of Santiago de Chile. Each participant in this research provided their written informed consent. This specified the tests that were carried out and also specified that the data obtained during the tests were used in this investigation. On the other hand, the document also specified that participants could withdraw from the research for no specific reason.

From these ECG segments, the representative model of the described population is obtained using instrumental variables for this. With the design of controllers and actuators, the biofeedback system shown in the block diagram in Figure 1 was modeled, designed, and implemented.

### 3.1. System Modeling

The modeling method of the dynamic system is based on the relationship between excitation (input) and response to the same (output). After confirming that these signals can be correlated, a representative model of HRV dynamic behavior in its transfer function form using the instrumental variables (IV) method.

To obtain and develop the input and output signals, 12 ECG samples are collected from individuals with the characteristics mentioned above. ECG segments have a period equal to *T* = 60 s with a sampling frequency of *f* = 360 Hz and a normal sinus rhythm. These segments were processed to obtain the HRV behavior of each individual.

HRV can be in relation to time defined as Equation (Equation 1) [17].
(1)$$RR_{peak}=\frac{\sum_{i=0}^{n}R\left(i\right)}{n}$$

If we assume from Equation (Equation 1) and Figure 2 that the R1 peak travels a Δx distance in Δt time, this implies that the peak has a *v* speed that in an ECG corresponds to HR [56].
(2)$$v_{RR}\left(t\right)=\frac{dx}{dt}$$

According to this analogy, the ratio of the speed change in the peak Ri can be obtained when this “travels” an *x* distance, that is, the acceleration of the heart rate shown in Equation (Equation 3). Then, in the form of HR acceleration, an HRV expression is obtained almost immediately, which does not depend on the time windows (T-period of analysis) but only on changes in two points or velocities (HR), which makes it different from the temporary HRV (Equation (Equation 1)).
(3)$$a_{RR}\left(t\right)=\frac{d^{2}x}{d^{2}t}$$

This action is carried out by the data acquisition block, specifically by the sensor, as seen in Figure 1.

The ECG collected is processed using the algorithm described in [56], obtaining the acceleration HR of each ECG.

#### 3.1.1. Input Signal

The input variable x(t) is a step function based on the HRV of the samples described above. The objective of this signal is to establish the level of HR acceleration, which is representative of HRV from the ECG samples of the population under study. The sample mean μ is taken as a sampling estimator of the data described, which requires a known probability distribution similar to its origin in all samples as its origin to be valid. This condition is tested through the Kolmogorov–Smirnov test [57,58]. Once confirmed that all samples originate from the same type of probability distribution, the step of λ mean is generated from these experimental data, with Equation (Equation 4) as the hypothesis.

**Hypothesis** **1.**
*$\hat{F}$ originates from an exponential distribution with parameters EXP(x,λ).*


**Hypothesis** **2.**
*$\hat{F}$ Does not originate from an exponential distribution with parameters EXP(x,λ).*


Taking this into account, the contrasting probability distribution is $F_{0}$.
(4)$$\hat{F}\left(x\right)=\left\{\begin{array}{c} 0\;\mathrm{for}\;x<X_{\left(1\right)}\\ \frac{i}{n}\;\mathrm{for}\;X_{\left(i\right)}⩽x⩽X_{\left(i+1\right)}\\ 1\;\mathrm{for}\;x⩾X_{\left(n\right)} \end{array}\right.$$

Here, the empirical and theoretical distributions $X_{i}\in \hat{F}$ and $x_{i}\in F_{0}$ are compared through Kolmogorov–Smirnov’s distance, obtaining the deviation of two empirical cumulative distributions, or a function of a reference theoretical cumulative distribution. This parameter was used because of its sensitivity to differences in location, as in the cumulative distribution function. Kolmogorov–Smirnov’s distance $D_{KS}$ is defined by Equation (Equation 5).
(5)$$D_{KS}\left(F_{0},\hat{F}\right)=\underset{\left(i=1,2,\ldots ,n\right)}{max}\left\{\left|F_{0}\left(x_{i}\right)-\frac{i}{n}\right|,\left|F_{0}\left(x_{i}\right)-\frac{i-1}{n}\right|\right\}$$

The function to counteract is composed of the vector $\hat{F}=a_{t}=\left[a_{1},a_{2},\ldots ,a12\right]$. This vector contains all ECG data from the population studied, as shown in Figure 3.

In the case of the theoretical or reference cumulative distribution $F_{0}$, it is estimated to be an exponential distribution in the form of x∼EXP(λ) for λ=0.0323 mm/s${}^{2}$.

Figure 4 represents the superposed accumulative distributions $\hat{F}$ for the experimental or empirical distribution and $F_{0}$ for the theoretical or reference distribution.

To accept the hypothesis $H_{0}$, the parameter $D_{KS}$ is set, which is the largest absolute difference between both distributions (theoretical and experimental) $\left(D_{KS}^{+},D_{KS}^{-}\right)$. The condition $D_{KS}\leq D_{\alpha }$ should be met according to Table 1. It is concluded that all the samples of experiment $\hat{F}=a_{t}$ originate from an exponential distribution of parameters x∼EXP(λ) and $D=max\{D_{KS}^{+},D_{KS}^{-}\}$ and a confidence interval α=0.05 [57,58].

Having proven that the sample distributions are equal and known, it can be stated that the mean is representative and, therefore, the value λ=0.0323 mm/s${}^{2}$ can be used as the maximum value of the step as an input signal (Equation (Equation 6)).
(6)$$x\left(t\right)=u\left(t\right)=\left\{\begin{array}{c} \lambda ,t\geq 0\\ t<0 \end{array}\right.\to \lambda =0.0323\;{\mathrm{mm}/s}^{2}$$

For a better understanding and reference of the results, millimeters (mm) are used as the measurement unit for length, since the normalized distance for the measurement and interpretation of ECG is the normal distance traveled by an R peak during 1 mm with respect to 0.04 ms [59].

#### 3.1.2. Output Signal

The λ of amplitude was obtained, and considering the duration of the different changes in HR acceleration from the collection of ECGs, an average threshold of the transient period is observed (tending to zero to maintain homeostasis [60]). Based on the average time of the ECG samples, in which the permanent regime period tends to zero, a response representative of the HR acceleration change process is established. This is shown in Figure 5, in which the HR acceleration data collected from the ECG superposed.

The twelve samples that form the HR acceleration vector $a_{t}$, are shown in Table 2, in which the average duration of the transient period before an HRV corresponds to Δt=4.5 ms since the acceleration $a_{2}$ is the most representative sample of the experiment and transient behavior of all samples, the signal y(t) was used as a response to the final model.

#### 3.1.3. Identification Model

Model identification is conducted to convert the model into a transference function. To this end, the instrumental variables method is used as an algorithm to determine the parameters of the sought model, employing the MSE (Mean Squared Error) index as a cost function to minimize error or differences between the responses of the sought model (y(t)) and the known response ($y_{m}\left(t\right)$). This is based on the adjustment IVs conduct in each parameter interaction (poles and zeros) of the sought transfer function, thereby identifying the dynamic model for HR acceleration in the studied population.

The input x(t) and output y(t) variables of the system are known and represent the excitation and response of the theoretical model; however, there is a causality relationship between this input signal x(t) and the output signal y(t). This allows the correlation between x(t) and y(t) based on the transfer function. The IV method uses the regression model given by Equation (Equation 7) [31,33,61]. Here, $\hat{y}\left(k\right)$ is a known parameter of the system and the output of the ϕ(k) system is a vector of m×n dimensions that have known parameters and are related to the input of the system, θ is an m×n vector of unknown parameters (transfer function). In turn, k∈N is the discrete sampling time of the variables under study.
(7)$$\hat{y}\left(k\right)=\varphi {\left(k\right)}^{T}\theta$$

With data from $\hat{y}\left(k\right)$ and φ(k) we attempt to minimize the cost function given by the least squares (Equation (Equation 8)), assessing $y_{(}i)$ from the known signal, and $y_{m}\left(i\right)$ corresponding to the output of the model under construction.
(8)$$v\left(k\right)=\frac{1}{k}\sum_{i=1}^{k}{\left[y\left(i\right)-y_{m}\left(i\right)\right]}^{2}$$

The estimation of the unknown parameter is conducted based on $\hat{\theta }$, which is replaced by the vector Φ, a set of parameters correlated to the regression of transfer function, i.e., the instrumental variables [61]. This means $\Phi^{T}=\left[\varphi \left(1\right),\varphi \left(2\right),\ldots ,\varphi \left(k\right)\right]$. Thus, the function to estimate the transfer function is defined in Equation (Equation 9).
(9)$$\hat{\theta }={\left[W^{T}\left(k\right)\Phi \left(k\right)\right]}^{-1}\left[W^{T}\left(k\right)Y\left(k\right)\right]$$
where $W\left(k\right)=\left[\hat{\varphi (1)},\hat{\varphi (2)},\ldots ,\hat{\varphi (k)}\right]$ are the regression parameters given by Equation (Equation 10)
(10)$$\hat{\varphi }\left(k\right)=\frac{\hat{B}\left(q^{-1}\right)}{\hat{A}\left(q^{-1}\right)}x\left(t\right)$$

The iterations executed to obtain the desired transfer function begin with a known transfer function form of the 1st and 2nd order, respectively. Shortening the transfer function parameter search based on the error yielded by $\epsilon \left(k\right)=y\left(t\right)-y_{m}\left(k\right)$. This process is summarized in Figure 6.

When applying the IV method in x(t) and y(t), transfer function H(s) is obtained, this function changes its parameters according to the values of a minimum MSE calculated based on the output of the desired system and the reference output, modifying the poles and zeros of the transfer function [31], which are contained in $\hat{\theta }$. As previously mentioned, to limit the search, the following two types of transfer functions were estimated.

Transfer function type 1: $H_{1}\left(s\right)=\frac{1}{s+a}$Transfer function type 2: $H_{2}\left(s\right)=\frac{s+a}{s^{s}+s+a}$

In this case, the results obtained for the transfer function are presented in Figure 7, in which the transfer function corresponds to Equation (Equation 11).
(11)$$\hat{\theta }=H\left(s\right)=\frac{22.95s+1.02}{s^{2}+2.57s+3.05}$$

Figure 7 shows the acceleration response signal of the known $y_{m}\left(t\right)$ and the response signal of the identified model y(t), These signals present a minimal error of *MSE* = 16.93% and a representativity of 83.03% with respect to the known signal $y_{m}\left(t\right)$.

#### 3.1.4. Controller Design

To design the controller, the MRAC controller is used through an adaptation mechanism that modifies the adaptation parameters of the PID controller based on a reference model [10]. The architecture in Figure 8 was considered for the development of the controller.

The controller corrects the deviation $e_{m}\left(t\right)$ associated with the previously obtained response of the model y(t) with respect to the output of the reference model $y_{r}\left(t\right)$. Based on the behavior of both outputs, the cost function *j* is minimized. In this way, the parameters of θ(t) are adjusted using the adaptation mechanism. The cost function is defined by the minimum integrative error of the output error expressed in Equation (Equation 12), in which $\gamma_{i}$ is the gain of the adaptation of the parameters $\theta_{i}$.
(12)$$j=\frac{1}{2}\int_{0}^{t}e_{m}^{2}\left(t\right)dt\to e_{m}\left(t\right)=y\left(t\right)-y_{r}\left(t\right)$$

Parameter adaptation is based on an MIT (Massachusetts Institute of Technology 1958) rule [62]. This seeks to rapidly reduce the error towards the gradient of the controller θ(t), parameters, i.e., these parameters are adjusted based on the reduction of $e_{m}\left(t\right)$ by means of the minimization of the cost function *j*. This is shown in Equation (Equation 13).
(13)$$\Delta \theta_{i}=-\gamma_{i}\left(\frac{\partial j}{\partial \theta_{i}}\right)$$

The temporary variation in the controller parameters $\Delta \theta_{i}$ with respect to their last adjustment, including the adaptation gain $\gamma_{i}$ is expressed in Equation (Equation 14).
(14)$$\frac{d\theta_{i}}{dt}=-\gamma_{i}\left(\frac{\partial j}{\partial \theta_{i}}\right)$$

This gain was selected to reduce the settling time and, at the same time, decrease the transition of the adaptation process of the adaptive system. By including the calculation of the gradient and adaptation parameters into the error of the responses $e_{m}\left(t\right)$, Equation (Equation 15) is obtained.
(15)$$\frac{d\theta_{i}}{dt}=-\gamma_{i}e_{m}\left(\frac{\partial e_{m}}{\partial \theta_{i}}\right)$$

As the output is a single variable, in which the gradient of the term $y_{m}\left(t\right)$ tends to zero, and as the reference model $y_{r}\left(t\right)$ does not depend on $\theta_{i}\left(t\right)$ the calculation of the tracking error gradient is reduced to the partial derivative of the heart rate acceleration model kG(s) in the closed loop with respect to the parameter of the control law $\theta_{i}\left(t\right)$, as shown in Equation (Equation 16).
(16)$$\begin{array}{c} \frac{de_{m}}{d\theta_{i}}=\frac{\left(\partial y-y_{r}\right)}{\partial \theta_{i}}=-\frac{\partial y_{r}}{\partial \theta_{i}} \end{array}$$
(17)$$\begin{array}{c} \frac{d\theta_{i}}{dt}=-\gamma_{i}e\left(\frac{\partial y_{r}}{\partial \theta_{i}}\right) \end{array}$$

Referencing these equations to the control signal *u* in Figure 8. The control law based on the adaptable parameters $\theta_{i}$ and the control signal or set point $u_{c}$ is defined according to Equation (Equation 18).
(18)$$u=\theta u_{c}$$

From this, these parameters are shown in terms of error in Equation (Equation 19).
(19)$$e_{m}\left(t\right)=y\left(t\right)-y_{r}\left(t\right)=kG\frac{d\theta_{i}u_{c}}{dt}-k_{0}G\frac{d\theta_{i}u_{c}}{dt}$$

If kG(s) is the heart rate acceleration (HRV) model, $k_{o}G\left(s\right)$ is the reference model with its corresponding gain. Calculating the sensitivity derivative $\frac{\partial y_{r}}{\partial \theta_{i}}$, indicates how the error changes according to the adaptation parameters θ. Its negativity indicates the minimization of the adaptation time that defines the robustness of the system (Equation (Equation 19)).
(20)$$\frac{\partial e}{\partial \theta }=kG_{s}\theta u_{c}=\frac{k_{0}}{k}y_{r}$$

The variation of the adaptation parameters over time can be defined using Equation (Equation 20).
(21)$$\frac{d\theta }{dt}=-\gamma_{i}\frac{k}{k_{0}}y_{r}e_{m}$$

Finally, the convergence point $k=k_{0}$ is obtained based on Equations (Equation 20) and (Equation 21).
(22)$$\frac{d\theta }{dt}=-\gamma_{i}y_{m}e$$

Given the parameters implemented in the controller, it is necessary to know the homologous reference level of the HRV model in the form of a transfer function (Equation Equation 11).

In the estimation of this model, the transfer function of the reference model $H_{r}\left(s\right)$ in Equation (Equation 23) is established using a settling time $t_{s}\leq 1s$ of and a maximum overshoot of $M_{p}<20\%$, and applying the criterion of 2% of $t_{s}$ [62].
(23)$$Hr\left(s\right)=\frac{183.1}{s^{2}+16s+183.1}$$

The controller handles 5V signal levels. Therefore, the general system is normalized in all the blocks of the system under study. The PID algorithm was adjusted by the Ziegler & Nichols reactive curve method [10], In this way, the following parameters were estimated in the controller: P=0.0138, I=0.241 and D=0.0012. Tuning this controller, the gain of the adaptation method can be empirically adjusted using $\gamma^{T}\left(i\right)=$ [0.1 3.3 6.6 9.9 13.3 20].

Figure 9a shows the response of the output of the system y(t), as well as the responses to different $\gamma^{T}\left(i\right)$ values.

Figure 9a shows several gains of the adaptation mechanism $\gamma_{i}$. One of these is chosen based on the energetic effort made by the controller when modulating the control signal in order to maintain the stability of the system. This is achieved with $\gamma_{i}=20$, obtaining a response that is controlled in terms of speed and overshoot of the system response. The above described is shown in Table 3 for the adaptation gains and their dynamic characteristics [62].

Figure 9b shows the response of the system y(t) with an overshoot superior to that of the reference model. However, this value is 16.77% and the settling time is 0.3604 s; therefore, it is an acceptable response according to the requirements established for the system response.

### 3.2. Actuator Design

The actuator of the system is based on light transduction; this means the conversion of the electric control signal into a light signal. In this case, the effect of light signals is considered within a spectrum range of visible light, as well as its effect on the behavior of individuals [63], being one of the first studies in the field. In this field, the works in [52,64,65] follow this line, demonstrating the psychological and physiological effect of light and its inherent properties, such as intensity and flickering frequency.

The actuator considers the work in [52,64] uses the effects of a short wavelength on the alertness and stress state of individuals. When using a short wavelength $\lambda_{o}=$ [380–450] nm, combined with flickering effects in the range of $f_{c}$ = [0–10] Hz; blue light affects the attention and alertness of individuals, which can become a conditioned reflex that allows maintaining attention during the performed activity, as well as regulating a “calmness” state that helps body homeostasis (regulation of heart rate acceleration or HRV to keep speed constant).

The control signal works within a range of $u_{c}$ = [0–5] v, This tension is converted into the $f_{c}$, frequency, which corresponds to the flickering frequency of the blue light in the actuator.

This voltage–frequency conversion is carried out via a VCO (Voltage Controlled Oscillator), which drives the continuous component of a modulated signal to vary in a linear and periodical way at the output. with a frequency that is directly related to the input of voltage v(t) that is modulated by the VCO. In this way, a response without time delay τ is obtained given the rapid response of the voltage–light intensity conversion to the responses of the dynamic system under study.

The VCO modulates the PWM signal response, which controls the overshoot current $i_{\gamma }$ of an LED diode of high brightness. After using Kirchhoff’s voltage laws to analyze the circuit in Figure 10 an experimental model is obtained that relates the overshoot current $i_{\gamma }$ to the relative light intensity of the LED light $i_{L}$ as shown in Equation (Equation 24).
(24)$$i_{d}=0.96e^{0.66i_{L}}$$

According to the circuit in Figure 10, the total voltage is $v_{t}=v_{d}+v_{l}$, as a function of current. This relationship is expressed as $v\left(t\right)=i_{t}\left(R_{d}\left(t\right)+R_{l}\right)$. The LED diode is not a linear load. Due to its low internal resistance, this parameter is not considered because an internal resistance $R_{in}\left(t\right)<0.01R_{th}$ is assumed to be 10% of the total circuit resistance. Therefore, the modulated current is the total current of the circuit, i.e., $i_{t}=i_{\gamma }$. However, the input voltage is modulated in PWM, which implies that the tension is not constant and that its variation is based on the modulation frequency $v_{t}=v\left(f\left(t\right)\right)$. In this way, a work cycle of $\%D=\frac{t_{1}}{t_{1}-t_{0}}$ is obtained.

The modulated voltage makes the amplitude of its continuous component vary based on the PWM frequency. Direct voltage (mean component) is the input signal to generate the conduction current $i_{\gamma }$ according to Equation (Equation 25).
(25)$$i_{t}=i_{\gamma }=i_{DC}=\frac{1}{\left(t_{1}-t_{0}\right)}\int_{0}^{t_{1}}v_{t}R_{l}$$

From Equation (Equation 25), and considering the signal amplitude of the constant PWM voltage and constant charge resistance $R_{l}$ constant, it is ensured that the direct current component is expressed according to Equation (Equation 26). This causes the desired effect on the brightness and frequency modulation of the LED diode, with %D = [0.001–0.0].
(26)$$i_{\gamma }=\frac{R_{l}}{\left(t_{1}-t_{0}\right)}\left(v_{t}t_{1}-v_{t}0\right)$$

Since the action of the actuator cannot interrupt the vision of individuals, the actuator was attached to lenses or safety glasses with specially designed adapters that cover the “monocular vision” spectrum, which is between $94^{\circ }$ and $110^{\circ }$ from the center of the left and right eyes, respectively. This is achieved using the diffusion angle of the LED in the mentioned adapters, as shown in Figure 11.

## 4. Results

The experimental setup was designed to validate the proposed HRV acceleration model, as well as testing and monitoring the functioning of HRV control through the design of the proposed controllers and actuators, thus, confirming the hypotheses of this study.

### 4.1. Experiment Design

The experiment was conducted using indoor cycling. This activity has been selected because it has characteristics of intense cardiovascular effort, which produces a high HRV variance according to the intensity changes that occur during the experiment. A magnetic induction roller is used, which generates a controlled force in the opposite direction to the traction of the bicycle’s back wheel. This implies a conversion of the torque generated by the biker’s legs to counter the opposite force generated by the roller, producing high physical wear that leads to high cardiovascular activity and, therefore, a high HRV. The specifications for the force generated by the roller are as follows.

Maximum generated torque: 12.2 Nm.Maximum braking force: 36 N.Mass inertia: 8 Kg.Maximum power: 700 W.

In this test, the HR of the individuals was monitored based on the duration of the R-R interval of the cardiac cycle [56], bicycle speed, algorithm occurrence time, and HRV over time. The last variable is expressed as the acceleration of HR with respect to the above parameters. The MRAC controller generated control actions in the light actuators to maintain HRV acceleration. The sensors and actuators installed on the individual are shown in the testbed scheme in Figure 12.

The room temperature was 25 °C and its relative humidity was 54%. Both variables were constant during the whole experiment.

The experiment was divided into four stages {A,B,C,D}:*A*: inertia output and generation of maximum effort in the least time.*B*: free run stage.*C*: controlled mode or activated HRV controller.*D*: free run stage.

The complete test had a duration period of *T* = 630 s. This period was divided into the stages mentioned above, as shown in Table 4.

The test was conducted with a different individual from those who participated in obtaining the model in Equation (Equation 11), but his physical characteristics belong to the universe of the obtained model, i.e., male, 42 years of age, the weight of 91 Kg and height of 173 cm.

### 4.2. Experimental Results

The results of the experiment are now presented, with a focus on the validation of the model and the study of the operation

These results were polished using the LOESS (locally estimated smoothed scatterplot) method, which uses linear regression by least squares to adjust models over their local data subsets, creating a function that describes the event. In this case, a 2% scope of data was used for all variables.

When presenting data that are statistically significant, it should be proven that the data have the same distribution; in this case, a normal distribution is validated with p>0.05 using the Kolmogorov–Smirnov method.

Figure 13 graphically shows the results obtained divided by each experimental stage in terms of time and stage duration.

The variables shown are as follows

$y_{exp}$ Response as acceleration of heart rate mm/s${}^{2}$.$y_{bpm}$ Response of heart rate as speed BPM.*v* Response of speed of the bicycle in km/h.

Analyzing the graphs in Figure 13a–c, and using the {A,B,C,D} segments of the experiment as a reference, we have the following: in segment *A*, there is an increase in all variables due to the inertia output of the bicycle, with a maximum effort that reaches the peak of HR acceleration. A maximum HR of 129 BPM, an HR acceleration of 0.0078 mm/s${}^{2}$, and a speed of approximately 20 km/h are reached. In segment B, the individual performs free cycling based on his physical resistance, achieving a maximum HR of 140 BPM, an HR acceleration of 0.009 mm/s${}^{2}$, and a maximum speed of 51 km/h. In segment *C*, with the active controller, it is observed that due to the light modulation, HRV is controlled; therefore, HR has a linear increase but an acceleration tending to zero, controlling the roller speed and consequently the applied effort. In this case, a maximum HR of 101 BPM is reached, as well as an HR acceleration of 0.009 mm/s${}^{2}$ and a speed peak in the last segment of 38 km/h. Finally, the *D* segment corresponds to free cycling.

Table 5 shows the descriptive statistics of the entire sample universe obtained in the experiment (including the results of all segments of the study). This aims to show a general summary of the variables under study, as well as the measurements of the data.

## 5. Analysis Result

After obtaining the results, two analyzes are conducted that compare the behavior of the segments in the experiment with the proposed model. The first model contrasts experimental and theoretical data, while the second one assesses controller performance during the experiment.

In this experiment, one of the main characteristics that can be seen corresponds to the high deviation that is obtained between the sample mean referring to standard deviation. These percentage differences found are $y_{exp}=18.41\%$ of HR acceleration, heart rate of $h_{exp}=16.48\%$, and roller speed of $v_{exp}=37.5\%$. This is mainly due to differences in the behavior of the variables for each of its segments.

### 5.1. Model Analysis

This analysis compares the time-based acceleration responses of the proposed HR with respect to the transient response obtained in the experiment. The experiment defined segment *A* for the transient study aimed at validating the model. In this segment, the individual emerges from inertia and reaches a maximum effort point. This translates into the effect on the response of HRV as HR acceleration and generated speed.

To establish the response differences between the proposed model y(t) and the response obtained experimentally $y_{exp}\left(t\right)$, the RMSE (Root Mean Square Error) index was used; this represents the absolute fit of two functions and has the property of being in the same units as the response variable. Lower values of RMSE fit the data vectors better. This index is a good measure of the precision with which the model predicts the response (evolution in t). In addition, the R${}^{2}$ fitness index and Pearson and Kendall correlation coefficients are used to compare the responses. The former is used to test the direct relationship of two continuous variables, while the latter is used to demonstrate the statistical dependence of two variables.

To test the model, the experimental response $y_{exp}\left(t\right)$ is considered in the time domain. In turn, the proposed model, as a transfer function, is in the frequency domain and therefore needs to transit to the time domain, so both responses belong to the same domain.

As the proposed theoretical model is a transfer function H(s), and X(s)=u(s) is an input signal (step type) in this system, it is feasible to obtain the response Y(s), based on Equations (Equation 6) and (Equation 11). It is then observed that the theoretical time response is developed according to Equation (Equation 27).
(27)$$\begin{array}{c} H\left(s\right)=\frac{29.95s+1.02}{s^{2}+s2.575s+3.05};\;\;\;\;\;\;u\left(s\right)=\frac{0.0323}{s}\\ Y\left(s\right)=H\left(s\right)X\left(s\right)=\left(\frac{29.95s+1.02}{s^{2}+s2.575s+3.05}\right)\frac{0.0323}{s} \end{array}$$

From this, Equation (Equation 28) is obtained.
(28)$$Y\left(s\right)=\frac{(29.95s+1.02)0.0325}{\left(s-(-1.2875+1.18i)(s-(-1.2875-1.18i)(s-0))\right)}$$

By applying Laplace’s inverse transform to function Y(s), it is assumed that this function can be considered a closed curve function and the poles of this function are singularities of the same. Then, the residue theorem can be applied to $L{\left(Y\left(s\right)\right)}^{-1}$. Consequently, the time response y(t) was obtained, which is necessary to compare this output with the experimental response $y_{exp}\left(t\right)$. The definition is presented in Equation (Equation 29).
(29)$$y\left(t\right)=L{\left(Y\left(s\right)\right)}^{-1}=\frac{1}{2\pi i}\oint_{C}^{\;}Y\left(s\right)e^{st}ds=\frac{1}{2\pi i}\left(2\pi i\sum_{k}\mathrm{Res}\left(Y\left(s\right),Z_{k}\right)\right)$$
where $\mathrm{Res}(Y\left(s\right),Z_{k})$ is the residue of the function Y(S) in the pole $Z_{k}$.
(30)$$\begin{array}{c} y\left(t\right)=L{\left(Y\left(s\right)\right)}^{-1}=\mathrm{Res}\left(Y\left(s\right),Z_{k}\right)/k=3;\\ y\left(t\right)=Y\left(s\right)e^{(-1.2875+1.18i)t}\left(s-\left(-1.2875+1.18i\right)\right)+\\ Y\left(s\right)e^{(-1.2875-1.18i)t}\left(s-\left(-1.2875-1.18i\right)\right)+\\ Y\left(s\right)e^{0t}\left(s-0\right) \end{array}$$
(31)$$y\left(t\right)=0.019-e^{\left(-1.2875t\right)}\cos\left(1.18t-\phi \right)$$

After calculating Equation (Equation 30), the response of the system y(t) is obtained (Equation (Equation 31)).

Knowing the temporary response of the HR acceleration model, the transient analysis period should be established. This period is defined as the point where the amplitude differences of both responses are null, as shown in Equation (Equation 32), where $y_{ss}\left(k\right)$ was the difference response.
(32)$$y_{ss}\left(k\right)=\sum_{i=0}^{k}\Delta y\left(i\right)=\sum_{i=0}^{k}\left(y\left(i\right)-y\left(i-1\right)\right)=0$$

Applying the condition to Equation (Equation 32) the transient period for the validation of the theoretical model y(t) with respect to the experimental data $y_{exp}\left(t\right)$ as *T* = 8.5 s.

Figure 14 shows the comparison between the signals y(t) and $y_{exp}\left(t\right)$ in relation to the transient period *T*. The experimental response for the HRV behavior in the *A* segment of the experiment has been obtained.

Table 6 shows the results of the assessment indices used in the analysis of the model.

The RMSE obtained from the comparison of the signals y(t) and $y_{exp}\left(t\right)$ was RMSE=4.35%, indicating the difference between the amplitudes of both functions and the periods of maximum energy consumed in the change of HR acceleration. This is good due to the statistical basis on which the HR acceleration model H(s) was developed in Equation (Equation 11), and represented in terms of time by y(t) in Equation (Equation 31). The fit of both curves, $R^{2}=94.65\%$ shows the coherence of directions and duration periods of the transient in both signals. Regarding the correlation, Pearson and Kendall ρ show a strong and positive correlation between both responses. Being a negative correlation, zero indicates no correlation and 1 pointing to a perfect correlation. In this case, both are higher than 60%, which implies a strong correlation and coherence of the responses, reinforced by a positive covariance. Therefore, the proposed HRV model is validated, as well as the method to obtain the same.

### 5.2. Controller and Actuators Analysis

In this section, the responses of the experiment’s segments *B* and *C* are compared. In segment *B*, the individual was active depending on his effort, trying to increase speed. Conversely, in segment *C*, the individual will try to increase the speed, but with the HRV controller activated.

The analysis consists of comparing the descriptive statistics with the experimental response in segment *B* with that in segment *C* to study the behavior of each response by segment, superposing them. The analysis is complemented by the study of the error over time e(t) of the responses. Table 7 shows the descriptive statistics of segments *B* and *C* of the experiment, including the calculation of the continuous component of each variable, which will be useful for the calculation of the error over time function e(t).

Figure 14 and Table 7 show that in segment *B*, the magnitude of the means with respect to HR acceleration is 91.1% larger than the magnitude of acceleration in the controlled segment *C*. This implies that the acceleration means used in segment *B* are much higher than in segment *C*, resulting in high dispersion in HR acceleration and consequently higher power dissipation. Regarding the *C* segment, minimal dispersion that reaches a deviation from the mean of only 3.54% is observed. Conversely, the dispersion of segment *B* corresponds to 7.2%, demonstrating the action of the controller and the regulation of HR acceleration through biofeedback.

The behavior of the error over time e(t) was individually calculated for the responses $y_{exp}\left(t\right)$ of the *B* segment and for $y_{exp}\left(t\right)$ of the *C* segment for a period equal to T=158s, which is the duration time of the controlled segment *C*.
(33)$$e\left(t\right)=\frac{|y_{exp}\left(t\right)|}{|y_{exp}\left(t\right)|-\left(\frac{1}{158}\int_{0}^{158}y_{exp}\left(t\right)dt\right)}$$

Equation (Equation 33) is the expression of e(t) where $y_{exp}$ are the independent experimental responses in segments *B* and *C*. The acceleration value of HR that represents the population under study, defined in Equation (Equation 6), corresponds to u(t)=0.00323 mm/s${}^{2}$, which is a reference value for the study of error behavior in experimental responses (value of reference variable).

The error behavior is shown in Figure 15, in which the responses $y_{exp}\left(t\right)$ for the segment *B* and *C*, as well as the reference variable u(t) superposed with one another. The difference between the behavior of the system response and the estimated reference variable u(t)=0.00323 mm/s${}^{2}$ is highlighted in this figure. The output of the *B*$y_{exp-B}$ segment exhibits a 4% difference with u(t). Likewise, $y_{exp-C}$, which is the output of segment *C*, presents a 15% difference with u(t), which is generated by variations of the mean of the reference variable (continuous) and the difference of the former with the reference variable u(t).

In segment *B*, the output has a sample mean of $\mu_{B}=0.079$ mm/s${}^{2}$ and a standard deviation of $\sigma_{B}=0.133$ mm/s${}^{2}$, with $\mu_{B}<\sigma_{B}$. This implies a high variance in the response of this segment compared to its principal component due to the absence of control mechanisms for the acceleration of HR (HRV) in this part of the experiment. However, segment *C* and its output have a sample mean of $\mu_{C}=0.016$ mm/s${}^{2}$ and a standard deviation of $\sigma_{C}=0.0114$ mm/s${}^{2}$, with $\mu_{C}>\sigma_{C}$. This results in low variance with respect to the principal component in the response of this segment, which is a consequence of the action of the controller on the system and reveals a change in the HRV response of the individual.

Another piece of evidence supporting the above corresponds to the difference between the means of each output, which corresponds to 79.7%. In addition, the difference between the standard deviations is practically 100%, i.e., in order of magnitude, the non-controlled segment *B* does not control the energy consumed, which implies that there is no control besides homeostatic control to maintain HR when experiencing signs of tiredness and fatigue. Instead, segment *C*, as it has lower variability in its central statistics, shows that the energy applied is much lower than that of segment *B* to increase HR speed slowly and linearly without sudden changes in HRV expressed as the acceleration of HR.

## 6. Conclusions

Through this research, a reliable and novel method has been presented for the estimation and identification of a dynamic model for the representation of HRV and its control, using light actuators and an MRAC-type control strategy.

We believe that this research is very valuable in the area of biofeedback, providing new mechanisms for the self-control of physiological variables such as HRV to avoid and improve the well-being and safety of individuals. Having the ability to use this technique as an anticipatory method of HRV behavior patterns under given conditions makes this research an important contribution to the area of engineering, medicine, and psychology.

From the results obtained, the behavior of the statistically obtained HRV model was compared with the estimated model based on the experimentally obtained data. This dynamic model represented in the form of a transfer function was subjected to reactive type tests such as the step response test. These responses were counteracted with the results applied to an individual not belonging to the initial group used for the system modeling, using a step test, where the individual emerged from inertia until a moment of stability with the maximum effort applied, obtaining a direct correlation $R^{2}=89.5\%$, verifying that the model is representative of a population group.

In reference to the self-control of the HRV, it is established that for a given test, the error in the time reached by using the controller and the signals from the light actuators is only 13.3% compared to the uncontrolled response that reached a dispersion close to 100%, verifying the effectiveness in the use of biofeedback through a continuous time controller with fast, noninvasive actuators such as light actuators.

## Figures and Tables

**Figure 1 sensors-22-07153-f001:**
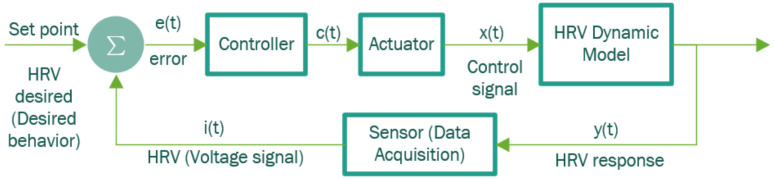
Biofeedback Control System.

**Figure 2 sensors-22-07153-f002:**
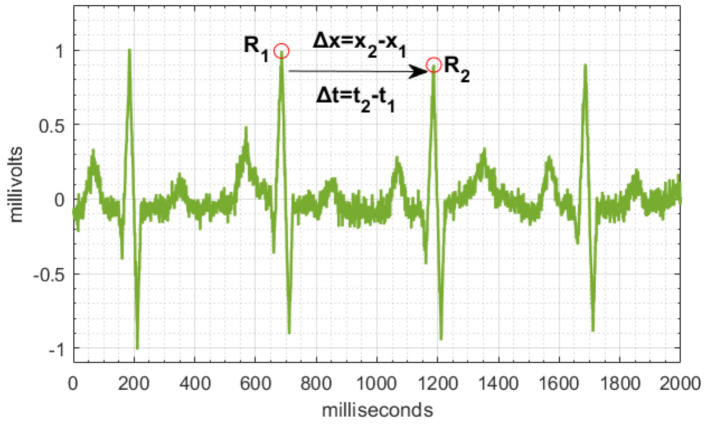
ECG processing algorithm.

**Figure 3 sensors-22-07153-f003:**
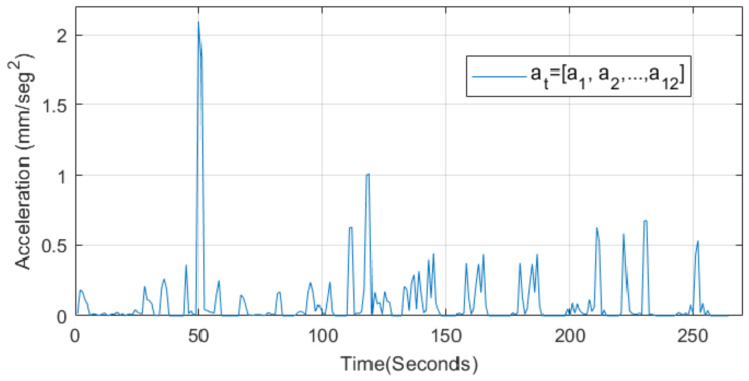
A vector with a collection of HR acceleration of each ECG.

**Figure 4 sensors-22-07153-f004:**
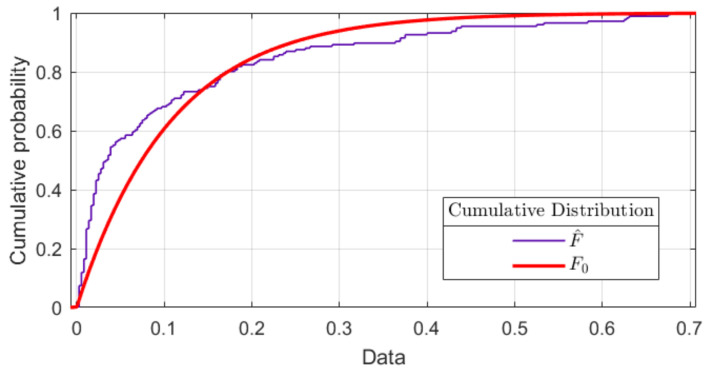
Superposition of reference theoretical $F_{0}$ and experimental $\hat{F}$ cumulative probability distributions.

**Figure 5 sensors-22-07153-f005:**
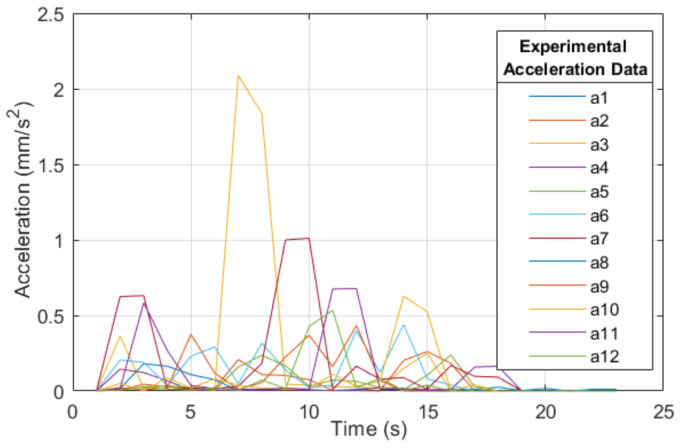
Superposition of samples of cardiac accelerations (experimental data).

**Figure 6 sensors-22-07153-f006:**
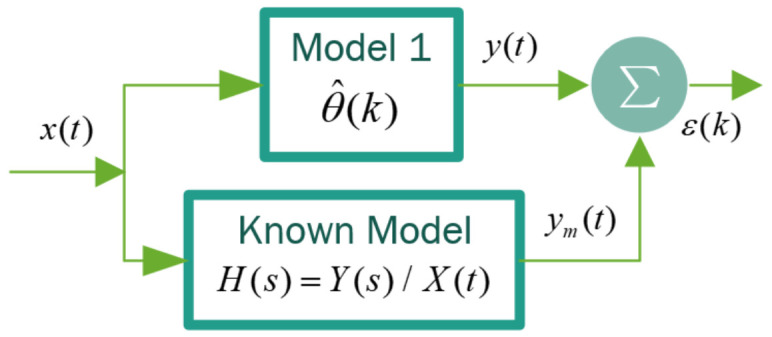
Application Instrumental Variable block diagram (experimental data).

**Figure 7 sensors-22-07153-f007:**
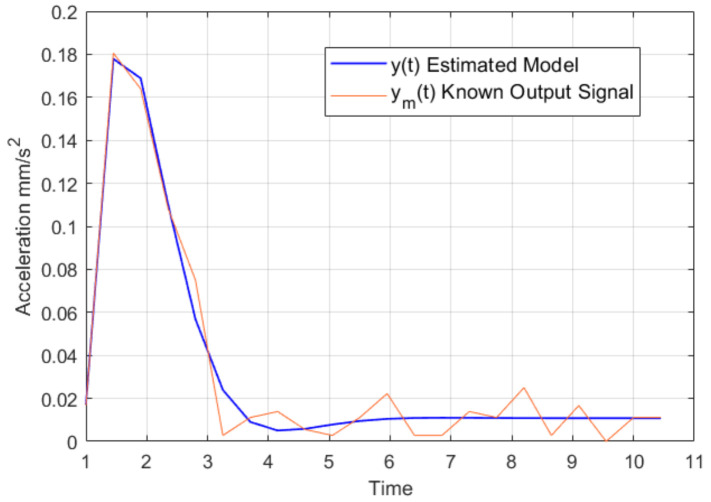
Graphic comparison of the reference model $\hat{y}$ and estimated model $\hat{\theta }$.

**Figure 8 sensors-22-07153-f008:**
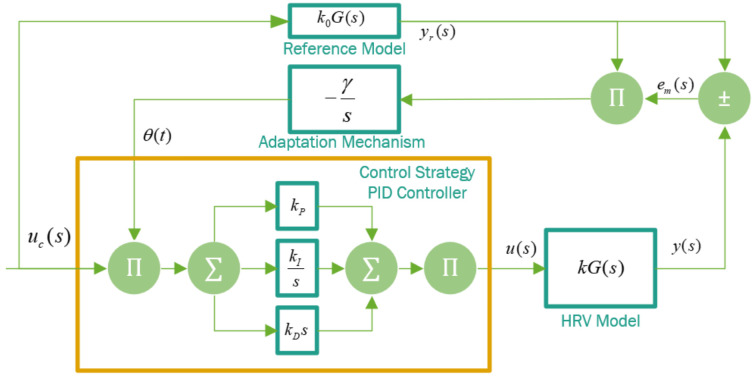
MRAC block diagram.

**Figure 9 sensors-22-07153-f009:**
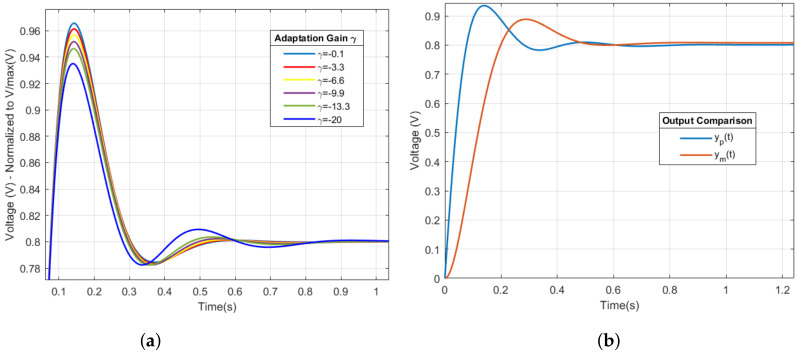
System output response and regulation. (**a**) Behavior of the adaptation mechanism $\gamma_{i}$. (**b**) Response of the HRV system y(t) respect to the reference model $y_{r}\left(t\right)$.

**Figure 10 sensors-22-07153-f010:**
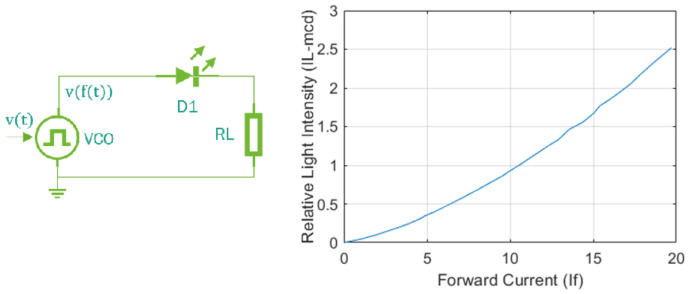
Base circuit to estimate brightness control and brightness intensity versus forward current graph.

**Figure 11 sensors-22-07153-f011:**
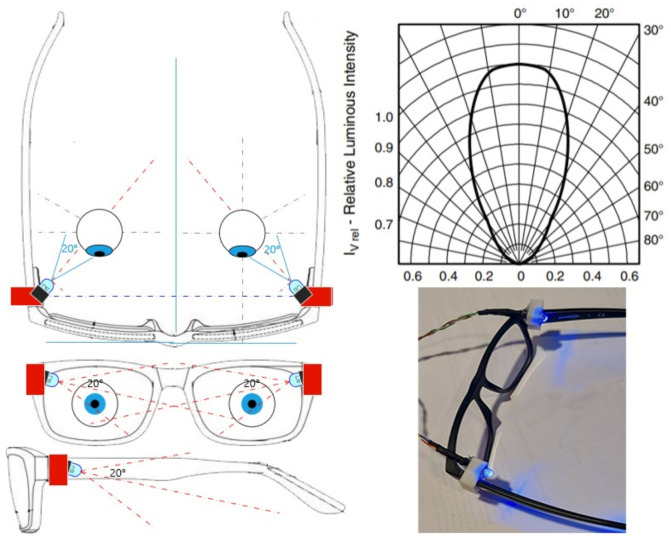
Actuator views, luminous intensity graph, and actuator implementation.

**Figure 12 sensors-22-07153-f012:**
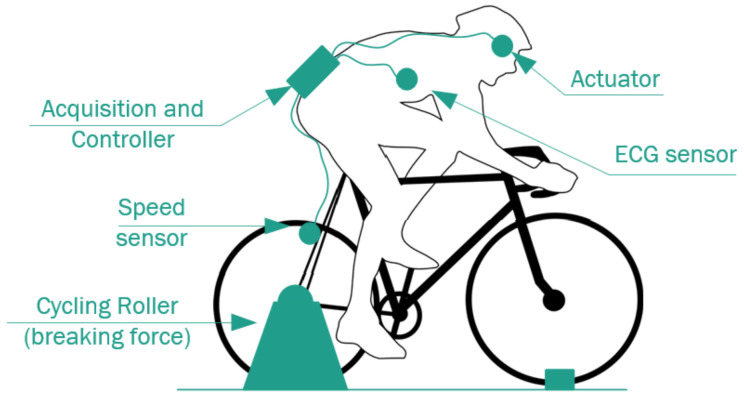
Testbed model validation.

**Figure 13 sensors-22-07153-f013:**
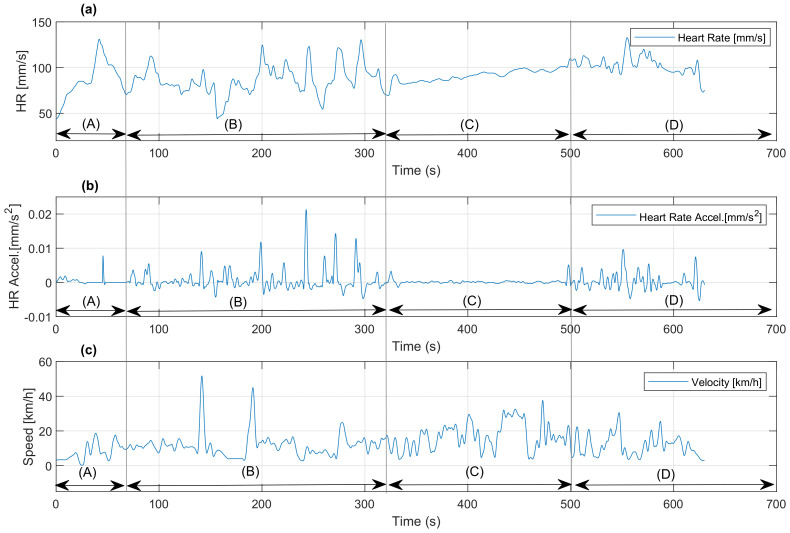
Experimental results, with the following variables: (**a**) HR in BPM, (**b**) HR acceleration mm/s${}^{2}$, and (**c**) speed km/h.

**Figure 14 sensors-22-07153-f014:**
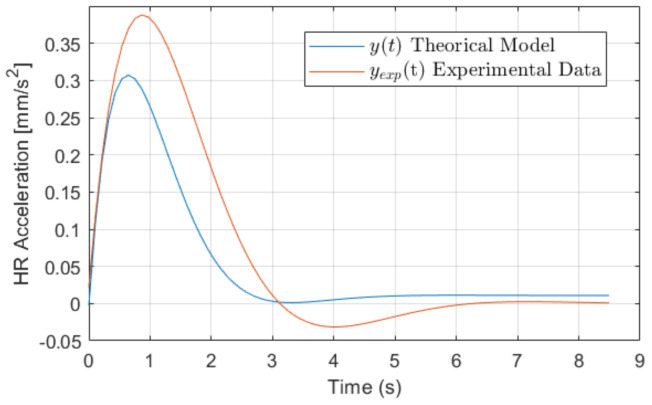
Response overlap plot.

**Figure 15 sensors-22-07153-f015:**
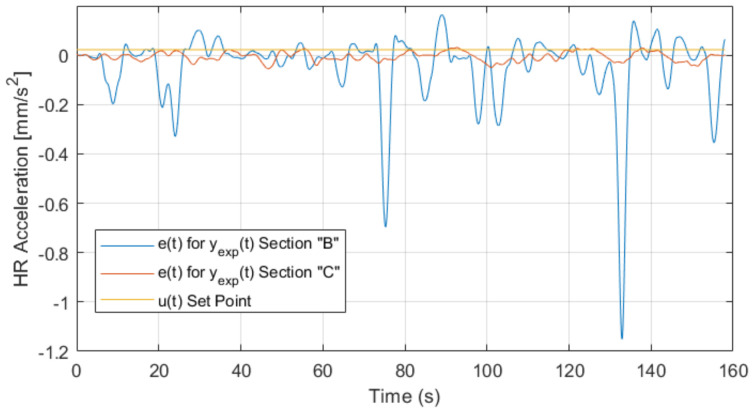
Error plot over time for $y_{exp_{B}}$, $y_{exp_{C}}$ and u(t).

**Table 1 sensors-22-07153-t001:** Hypothesis test result.

Experimental Distribution		
**Parameter**	**Magnitud**	**Unit**
λ	0.0323	mm/s${}^{2}$
$D_{KS}^{+}$	0.04392	mm/s${}^{2}$
$D_{KS}^{-}$	0.04378	mm/s${}^{2}$
$D^{\alpha }$	0.04392	mm/s${}^{2}$

**Table 2 sensors-22-07153-t002:** Duration of transient time per sample ($a_{i}$).

Sample	a1	a2	a3	a4	a5	a6	a7	a8	a9	a10	a11	a12
$t_{0}$ [s]	1	11	5	15	5	6	3	2	11	7	5	10
$f_{f}$ [s]	6	16	9	18	9	10	9	6	15	11	10	16
Δt [s]	5	5	4	3	4	4	6	4	4	4	5	6

**Table 3 sensors-22-07153-t003:** Response variation as a function of $\gamma_{i}$.

Adaptation Gain $\gamma_{i}$	Max Overshoot %	Settling Time $t_{s}$	Rising Time $t_{r}$
0.1	20.7316	0.2801	0.0579
3.33	20.5850	0.279	0.0579
66	19.895	0.386	0.0580
9.95	18.944	0.388	0.0582
20	16.771	0.36	0.0588

**Table 4 sensors-22-07153-t004:** Hypothesis test result.

Stage	Description	Period *S*	Remarks
A	Step test	68	Initial movement from repose.
B	Free run	252	Freecycling, aiming to reach the max. HR.
C	Controller activated	246	Controlled cycling, aiming to increase. the speed while having the HR acceleration constant (control action).
D	Free Cycling	64	Freecycling, aiming to reach the max. HR.

**Table 5 sensors-22-07153-t005:** Descriptive statistics of the experimental results.

Variable	Min	Max	Mean	Std. Dev.	Variance
HR Acceleration ($y_{exp}$) [mm/s${}^{2}$]	−0.0228	0.09119	0.0019	0.01032	0.000107
HR ($h_{exp}$) [BPM]	43	131	91	15	211
Speed ($v_{exp}$) [km/h]	2.2	29.6	12.8	4.8	22.7

**Table 6 sensors-22-07153-t006:** Statistical fit and correlation parameters.

RMSE	R${}^{2}$	ρ Pearson	ρ Spearman	Covariance
0.0435	0.895	0.6203	0.9465	0.0180

**Table 7 sensors-22-07153-t007:** Statistical parameters for sections “B” and “C”.

Section	Variable	Mean	Std. Dev.	Var	Vdc
B	Acceleration HR [mm/s${}^{2}$]	0.000691	0.003087	0.000010	0.022161
B	HR [BPM]	88	15	238	2821
B	Speed [km/h]	27.4	8.2	66.4	877.3
C	Acceleration HR [mm/s${}^{2}$]	0.000063	0.000223	0.000000050	0.001171
C	HR [BPM]	94	6	30	1756
C	Speed [km/h]	41.2	10.7	115	771

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
