# Peer review of "Heart Rate Variability Control Using a Biofeedback and Wearable System"

_sensors, 2022, doi:10.3390/s22197153_

Round 1
Reviewer 1 Report
I made many comments in the PDF. In summary the introduction does not justify why a new HRV parameter is needed, what is the benefit? No clear aim given. The materials and methods mainly written in future tense like in a research proposal. The research involved humans, but the number of individuals are not reported. This human experiment is not approved by any ethical committee. The results section contains some data example, but not real result tables. The discussion part contain not any references, than how it is discussed? Conclusions are too long, kind of summary, no clear conclusion, take home message. Still not clear that how firefighters come into the scene?

Author Response
"Please see the attachment."

Reviewer 2 Report
The study described in this manuscript is an interesting approach to modelling heart rate variability in physically active people. However, the descriptions are sometimes hard to understand. That hampers the potential replication of the presented steps and interpreting the results.
Major points:
1. The purpose of your study was not explicitly stated in the introduction - it is mentioned briefly in the introduction and in the conclusions in a more detailed description.
2. Why do you use the future tense for describing the experiment? As a potential reader, I expect the description of an experiment that was conducted before the submission of the paper.
3. I would recommend proofreading the manuscript to improve the legibility.
4. Your study involves human subjects. There is no mention of the approval of your experiment by a bioethics committee. Was your study approved by the bioethics committee or it did not apply in your case? If so, please state the reason.
5. Figure 13 (b): the label of the y-axis is not consistent with the label in the legend. What is the unit of velocity in this graph: m/s² or mm/ms²? Or maybe mm/s²?
6. Figure 13 (c): the label of the y-axis is not consistent with the label in the legend. What is the unit of velocity in this graph: m/s or km/h?
7. I would recommend comparing the obtained model with other existing models of heart rate variability.
Minor points:
Line 27: "Entre otros" could be translated.
Lines 126-131: correct the formatting and replace the parentheses "()" with a colon placed after the name of quantity, e.g. Age: 35-45 years.
Line 127: "Wight" could be replaced with "Weight"
Lines 164, 172, 178, 402, : inconsistent spelling of "Kolmogorov-Smirnov"
Line 208-209: Did you mean "transfer function" for TF?
Lines 372-375: see the comment for lines 126-131.
Author Response
"Please see the attachment."

Round 2
Reviewer 1 Report
The manuscript improved a lot. Now it is possible to understand the aims. What I still miss is some information regarding the authorization of the human study, even if it is non invasive.
line 359 word repetition
line 423 table 5 shows only one segments (which?) descriptive statistics for the three parameter. If you had 4 segments, 12 individuals, than a statistical analyses should be performed. From the graphical examples that should clearly prove the effectiveness of the controller.
line 430 include the values in table 5 for easy control
Reviewer 2 Report
The new version has been vastly improved compared to the previous one. However, there are a few suggestions before accepting for publication:
1. I would also mention the bioethics approval in section 3, after the description of subjects.
2. Paragraphs 2-3 (lines 25-29) could be merged into one paragraph.
3. Lines 47-48: “Latter translates into problems of fatigue, dizziness, weakness, fainting, and so on (...)” could be paraphrased as a simpler and more specific sentence.
4. Line 359: unnecessary “o validate”.
5. Section 5 could be renamed “Analysis of Results”.
